# Complement System Inhibitory Drugs in a Zebrafish (*Danio rerio*) Model: Computational Modeling

**DOI:** 10.3390/ijms241813895

**Published:** 2023-09-09

**Authors:** Dayanne Carla Fernandes, Denise V. Tambourgi

**Affiliations:** Immunochemistry Laboratory, Butantan Institute, São Paulo 05503-900, Brazil; dayanne.fernandes.esib@esib.butantan.gov.br

**Keywords:** Cp40, PMX205, C3a, C5a, C5aR1

## Abstract

The dysregulation of complement system activation usually results in acute or chronic inflammation and can contribute to the development of various diseases. Although the activation of complement pathways is essential for innate defense, exacerbated activity of this system may be harmful to the host. Thus, drugs with the potential to inhibit the activation of the complement system may be important tools in therapy for diseases associated with complement system activation. The synthetic peptides Cp40 and PMX205 can be highlighted in this regard, given that they selectively inhibit the C3 and block the C5a receptor (C5aR1), respectively. The zebrafish (*Danio rerio*) is a robust model for studying the complement system. The aim of the present study was to use in silico computational modeling to investigate the hypothesis that these complement system inhibitor peptides interact with their target molecules in zebrafish, for subsequent in vivo validation. For this, we analyzed molecular docking interactions between peptides and target molecules. Our study demonstrated that Cp40 and the cyclic peptide PMX205 have positive interactions with their respective zebrafish targets, thus suggesting that zebrafish can be used as an animal model for therapeutic studies on these inhibitors.

## 1. Introduction

The complement system is governed by a cascade of enzymatic events that provide hosts with one of the first mechanisms for innate immune defense. It is composed of over fifty blood-circulating and cell-surface-expressed proteins that participate in the recognition and clearance of invading pathogens. The activation of the complement system can occur through three pathways, classical, lectin and alternative, all of which converge for the cleavage of the central complement component C3 through the action of C3 convertases [1,2]. The cleavage of the C3 component leads to the generation of two fragments, C3b and the anaphylatoxin C3a. C3b is involved in the formation of C5-convertase which, in turn, cleaves C5 into C5b and the anaphylatoxin C5a. C5b interacts with C6, C7, C8 and several C9 proteins to form the membrane attack complex (C5b-9n or MAC), which generates a lytic pore on the target membrane. The anaphylatoxins C3a and C5a are potent pro-inflammatory mediators, via their interactions with specific receptors such as C3aR and C5aR1 [2]. Thus, the complement system is an important contributor and amplificatory mechanism for inflammation if activated in excess or if inappropriately controlled.

The complement system is involved in the pathogenesis and clinical manifestations of several systemic diseases, such as systemic lupus erythematosus (SLE) [3,4], vasculitis [5], antiphospholipid antibody syndrome [6], systemic sclerosis [7], dermatomyositis [8], rheumatoid arthritis [9], AMD [10], Alzheimer’s disease [11] and asthma [12], among others. The large number of pathological conditions in which the complement system is involved has stimulated the development of therapeutic interventions [13]. Thus, various components of this system, such as the anaphylatoxins and their receptors, have been considered promising therapeutic targets in inflammatory diseases [14,15].

The zebrafish (*Danio rerio*) is a model for studies relating to immunology, pharmacology, toxicology, cancer, neurodegenerative diseases, inflammation and other conditions [16,17,18,19,20,21,22]. It forms a versatile model in research terms and its use is well disseminated, given that its maintenance and reproduction are not complex and that its body is transparent during its initial development [23]. Moreover, the costs of zebrafish maintenance are advantageous in comparison with rodents; implementation of transgenic models is possible; and the growth and development of zebrafish are rapid [24,25,26]. The zebrafish complement system is structurally and functionally similar to that of humans, and these fish express homologs for all of the fundamental mammalian complement components [27].

As mentioned above, although the complement system forms a means of defense for hosts, exacerbated responses may lead to a change from a homeostatic state to a physiopathological state, thus resulting in severe immunological and inflammatory disorders [2,28]. One alternative for controlling the inflammatory process would be to use specific inhibitors of the complement system and its respective receptors and molecules, such as synthetic peptides [29,30] with neutralizing or modulatory action. Among the synthetic peptides with the greatest potential for inhibiting the complement system are Cp40, which inhibits conversion of C3 to C3b [31,32], and the second generation of PMX53, i.e., PMX205, which selectively blocks receptor 1 of C5a [33].

The analog peptide of compstatin Cp40 (Tyr(D)-Ile-[Cys-Val-Trp(Me)-Gln-Asp-Trp-Sar-Ala-His-Arg-Cys]-mIle-NH2) presents a high affinity to the human C3 molecule. Through its binding to intact C3 and/or the C3b fragment, it is capable of inhibiting the activation of the complement system [34]. Cp40 has been considered to be a potential therapeutic agent in models for various diseases of inflammatory or autoimmune nature, including sepsis [35,36], periodontitis [37], envenomation [38,39], hemorrhagic shock [40], nocturnal paroxystic hemoglobinuria [31] and autoimmune anemia [41]. Most recently, it has been demonstrated that it is capable of improving survival and reducing hypoxia in patients with severe COVID-19 [42]. 

PMX205 (N(2)-(3-phenylpropanoyl)-L-Orn-L-Pro-3-D-Cha-L-Trp-L-Arg) is a peptide antagonist to C5aR1. This molecule has been used as a pharmacological tool and therapeutic agent in experimental models for inflammatory diseases, including amyotrophic lateral sclerosis [43], colitis [44], meningococcal meningitis [45], Alzheimer’s disease [46], envenomation [38,47] and periodontitis [48].

The interaction between a drug and its receptor can be assessed at an early stage through in silico studies to provide a better understanding of the mechanism of action of this drug, prevent side effects and contribute towards the development of next-generation drugs [49,50]. In the present study, we used computational approaches together with bioactivity databases in order to assess predictions for Cp40 and PMX205 in relation to the C3 molecule and the C5aR1 receptor of the zebrafish complement system. Thus, we aimed to predict the 3D structure of the peptides and their biochemical characteristics, their target-protein binding and docking properties and the dynamics of competition or interaction of the protein/receptor complex. Our results provide a new perspective for the use of the zebrafish model for studying the activation and inhibition of the complement system and also suggest that, besides PMX205, Cp40 may also be used as a complement inhibitor in the zebrafish model.

## 2. Results

### 2.1. Analysis on the Homology of C3 Molecules in D. rerio and H. sapiens 

The mature C3 molecule is composed of two chains, α and β. This component possesses the same composition of domains and the same number of chains in both *D. rerio* and *H. sapiens*. Figure 1 shows the alignment of the amino acid sequences of the β chain of the C3 component from humans and zebrafish. The β chain of the C3 molecules from these two species present 63.7% similarity and 43.9% identity in their amino acid sequences.

### 2.2. Comparative Structural Analysis on the C3 Molecules in D. rerio and H. sapiens

Figure 2a,b shows a comparative analysis of the C3 β-chain proteoforms from *H. sapiens* and *D. rerio*; the presence of similar amino acid sequences is marked in blue. This comparative evaluation of the physicochemical properties of the C3 β-chain MG4 and MG5 domains between the species revealed that hydrophobic areas were predominant in the *H. sapiens* molecule, while positive and negative areas were more abundant in *D. rerio* (Figure 2c). The area of the binding site shows sets of hydrophobic amino acids, heterogeneously distributed between the species, and the same was observed for positive and negative charges. The values of the number of amino acids and their areas of occupation are shown in detail (Figure 2d). However, when we evaluated the networks formed by hydrogen bonds, the simulation of the MG4-5 domains of *D. rerio* showed nine networks in the region of the Cp40 binding site, versus seven in *H. sapiens* (Figure 2e). In detail, neutral charge was found to predominate on the Cp40 binding surface of the two proteins. On the other hand, hydrophobicity was heterogeneously distributed between the two proteins, with a predominance of hydrophilic areas (Figure 2f). Lastly, when Cp40 was evaluated, a predominance of hydrophobic areas and areas isolated from positive and negative charges was observed (Figure 2g).

### 2.3. Docking-Based Screening of C3 and Cp40 Interactions in D. rerio

The first virtual screening to determine specific and non-specific interactions, based on cavitation and blind docking, generated ten interactions or models (Figure 3a) of which only Model 8 of the *D. rerio* C3–Cp40 interaction was specific or close to the model-based binding site (PDB ID: 2QKI), while Models 7 and 10 for the *H. sapiens* interaction were close to the binding site (Figure 3b). Next, upon confirming the possibility of interaction between Cp40 and C3 from *D. rerio*, a second docking using a hybrid fitting strategy was performed, which provided input data and directed the interactions to the receptor-binding amino acid residues generated in Model 8#. This model presented the central location coordinates (x, y, z) closest to the binding site (PDB ID: 2QKI) (Figure 3c) and the chosen residues are shown in detail, in association with cavitation/pockets between the MG4 and MG5 domains (Figure 3(c.1)). Analysis on the directed docking showed that all interactions were directed towards the activity site area in *D. rerio* C3 (Figure 3d). Model 2 (Figure 3e) showed considerable values for the docking score (−116.71) and confidence score (0.3394). Moreover, the flexibility of this connection (Figure 3(d.1)) was closest to Model 0 and the crystallographic model (PDB ID: 2QKI). Redocking reshaped Template 2 and adjusted the peptide flexibility to improve fitting and similarity with the crystallographic template.

### 2.4. In Silico Analysis on the Interaction of Cp40 with the C3 Molecule in D. rerio

For the docking analysis, the β chain of the C3 molecule from *D. rerio* was constructed (Figure 4(a,a.1)). Our results showed that Cp40 binds to the β ring of the C3 β chain of *D. rerio*, located between domains MG4 and MG5 (Figure 4(a.1)), a position similar to that of its interaction with the human C3 molecule. 

In the C3 structure of *D. rerio*, Cp40 interacted with a smaller part of the β ring (of around 10 aa), compared with the interaction of the inhibitor with the human C3 (Figure 4(b.1,c.1)). In the redocking of the complex, i.e., β chain/Cp40, this difference was corrected and a greater number of amino acids of the β ring seemed to interact with Cp40 in *D. rerio* (Figure 4(d.1,d.2)). Figure 4(d.3) shows the quadratic deviation measurements (RMSDs) of the mean distance between the overlapping protein atoms for each of the 200 simulated models, and the values are expressed in msBB vs. score.

Overall, the interaction between the β chain of *D. rerio* C3 and Cp40 (Figure 4e) includes three strong hydrogen bonds, among the glycine residues GLY-336, GLY-445 and asparagine ASN-446 of C3, with the respective residues of Cp40, glutamine GLN-5, histidine HIS-10 and isoleucine ILE-1. In *H. sapiens* (Figure 4f), three hydrogen bonds also form between the β chain and Cp40: threonine THR-385, GLY-339 and arginine ARG-450. These residues interact with threonine THR-4, valine VAL-3 and glutamine GLN-5 of the binder. However, the three residues of the C3 of *D. rerio* present stronger hydrogen interactions with Cp40 than those of the binder to the human molecule.

### 2.5. Analysis of the Homology of the C5aR1 Molecules in D. rerio and H. sapiens

Figure 5 shows the alignment of the C5aR1 molecules of humans and *D. rerio.* They present 56.9% similarity and 39.1% identity, in comparison to using the FASTA sequence.

### 2.6. Comparative Structural Analysis on C5aR1 in D. rerio and H. sapiens 

The proteoform of the C5aR1 molecule from *D. rerio* presents a composition of seven transmembrane loops (Figure 6a,b). The C5aR1 molecules from *D. rerio* and *H. sapiens* (Figure 6c) predominantly present a neutral charge, with a hydrophobicity region in the intramembrane portion. In the extra and intracellular portions of the protein, we observed hydrophilic regions and mixed positive, negative and neutral charges. Analysis on the cavitation of C5aR1, in the two species, revealed a main site for binding with PMX205 (Figure 6(c.1)). *D. rerio* presents a pocket of greater volume that suggests that it has greater flexibility for interaction with the binding peptides (Figure 6d). In the comparative analysis on C5aR1, the distribution of hydrophobic areas and areas with positive charges was greater in C5aR1 of *D. rerio*, while areas with negative charges were greater in *H. sapiens* (Figure 6e,f). Analysis and comparison of the distribution of networks formed by hydrogen bonds in C5aR1 of *H. sapiens* and *D. rerio* showed that two networks were identified in both species. Network 1 of the two species is located at the same coordinates, and network 2 at different coordinates, but both are located at the binding site (Figure 6g).

### 2.7. Docking-Based Screening of C5aR1 and PMX205 Interactions in D. rerio

The modeling of PMX205-C5aR1 followed the same path, and the first screening based on cavitation and blind docking generated ten interactions or models for *D. rerio* (Figure 7a) and *H. sapiens* (Figure 7b). Among the ten models generated, only Model 1# for *D. rerio* presented an association with the interaction site based on the well-resolved model (6C1R), while for *H. sapiens* associations were shown in models 1#, 4 and 7. Model 1#, in common between the two species, demonstrated close cavitation and location values (center x, y, z) (Figure 7c), thus directing the amino acid residues of Model 1# for use as input data for hybrid fitting. The result from this modeling generated ten models for each species and, out of these ten interactions for *D. rerio*, eight models (1–7 and 10) were located at the binding site based on the crystallographic model and two models (8–9) at nonspecific sites (Figure 7d), whereas for *H. sapiens* all models were directed to the binding site (Figure 7e). Comparison between the species showed that the values were close (Figure 7f). However, even though Model 1 for *D. rerio* presented the best docking score (−200.90) and confidence score (0.7346) through analysis using the docking location criterion, which was close to the crystallographic model (6C1R), Models 6# and 7# stood out (Figure 7f) and were directed towards redocking. Hence, the model of choice was Model 6#.

### 2.8. In Silico Analysis of the Interaction of the Inhibitor PMX205 with C5aR1 in Danio rerio and H. sapiens

The peptide PMX205 is shown in its 3D form in Figure 8a. The interaction of the PMX205–C5aR1 complex in *D. rerio* occurred at the same site as this interaction in humans (Figure 8I). The docking analysis on the binder–receptor interaction for the two species is shown in Figure 8(c.1,d.1). The details of the peptide interaction with C5aR1 are presented in Figure 8(c.2,d.2). This shows that there was better occupational coupling of PMX205 to the pocket of C5aR1 in *D. rerio*. The main interactions between the amino acid residues of the receptors of *D. rerio* and *H. sapiens* and the binder, based on hydrogen bonds, are shown in detail in Figure 8(c.3,d.3). 

## 3. Discussion

The zebrafish model has been considered for studying the complement system since some of the components involved in the activity and regulation of the pathways of this system have been identified and cloned, including the proteins C1q [51], C4 [52], factor B [53], C3 [54], MBL [55], factor H [56], properdin [57] and CD59 [58]. Other regulatory factors of the complement system have been observed in studies on the induction of inflammatory responses in zebrafish [59]. Furthermore, high-resolution genome mapping and comparisons with the human genome have shown that the human genome has a similarity of approximately 70% with orthologous genes of zebrafish, thus emphasizing the biological reliability of this model [60].

Various studies have demonstrated that the use of inhibitors of the complement system is a promising therapeutic strategy for curbing the persistent uncontrolled activation of this system. Such activation contributes to the development of various pathological conditions [61]. Cp40, a member of the compstatin family, is one of the promising inhibitors of the complement system, since it binds to the central component of the cascade, the C3 molecule, and blocks its cleavage with C3 convertases [62]. While Cp40 in humans and non-human primates shows high levels of inhibitory activity in relation to the complement system, it has been shown that in mice, rats, rabbits, dogs and pigs, compstatin analogues do not present any notable inhibition of activation of the system [63]. On the other hand, Cp40 action in the zebrafish model remains obscure. Given that the binding region of Cp40 has been identified in the C-terminal portion of the β chain of C3 [64], and that zebrafish possess conserved complement molecular pathways compared with humans [65], the findings from the present study throw light on the hypothesis of the possible interaction between Cp40 and C3b in zebrafish.

Virtual prediction of protein structures using artificial intelligence has been well postulated in the literature and important advances have been made with regard not only to prediction and protein structure studies [66,67], but also to structure refinement [68], topology [69] and virtual docking interaction [70]. Thus, in the present study, servers and platforms that already existed were used. We found that these were highly reliable for predicting the protein structure, without any need for a new programming language to perform the analyses. We are aware of the limitations of in silico studies and we therefore intend to test these complement inhibitors in vivo in further studies, using zebrafish as a model for diseases associated with complement system disorders.

The present hypothesis can be initially supported through comparative structural analysis on the C3 molecule between *H. sapiens* and *D. rerio*. The mature human C3 molecule is composed of two chains, β (residuals 1–645) and α (residuals 650–1641), which together form thirteen domains, i.e., eight macroglobulin domains (MG1-8), one linker domain (LNK), one anaphylatoxin domain (ANATO), one CUB domain, one domain containing thioester (TED) and one C345C domain [71]. The C3 components of *D. rerio* have the same domain composition and the same number of chains. Furthermore, the similarity in the amino acid composition and structural identity of the β chain of C3 molecules in *H. sapiens* and *D. rerio* strengthens the possibility that peptide–protein interactions may indeed be possible (Figure 1 and Figure 2a,b).

Another promising complement inhibitor is the cyclic peptide PMX205, a potent C5aR1 antagonist [33]. The human C5aR1 molecule is composed of 350 amino acids [72]. The activation of C5aR1 in humans requires the binding of C5a to two distinct sites: the main one, which is located in the extracellular N-terminal portion (amino acids 2-22), and another signal-transducing site, which is formed by the extracellular portions of α-helices III, VI and VII [73,74,75,76].

Structural analysis on C5aR1 in *D. rerio* showed a structural similarity to the human homologous receptor and some amino acid sequences paired with this receptor (Figure 5 and Figure 6a,b). The general structure of human C5aR1 consists of a canonical helical arrangement of seven transmembranes (TM1–TM7) and an orthosteric site for class A G-protein-coupled receptors. Another important piece of evidence comes from a model based on the 6C1R crystallographic model, in which the interaction of PMX53 linked to the orthosteric pocket of C5aR1 formed a structure similar to the β strand [77]. This evidence, together with the docking results, suggests that the interaction between PMX205 and C5aR1 in *D. rerio*, located in the same orthosteric binding pocket as PMX53, occurs similarly to the interaction seen in the human model.

The composition of protein surfaces determines both the affinity and the specificity of protein–protein interactions. Moreover, the matching of hydrophobic contacts and charged groups at the two interface sites is crucial for ensuring specificity [78]. Among these, hydrophobic interaction seems to have an important effect on the molecular interaction between Cp40 and the zebrafish C3 molecule. Although the total hydrophobic areas of the MG4 and MG5 domains of *H. sapiens* are more abundant, considerable numbers of hydrophobic amino acid chains are present on the surface of the Cp40 binding site in zebrafish (Figure 2c,d). Furthermore, all of the amino acids of Cp40 present a hydrophobic characteristic that can facilitate molecular interaction (Figure 2g). Shekhawat et al. (2022) [79] used docking analysis on human RBD-ACE2 protein interaction to show the importance of hydrophobic interaction, and the same was demonstrated in our results.

The most abundant distribution of positive and negative charges on the surface of the MG4 and MG5 domains was on C3 in *D. rerio* (Figure 2c,d,f). This suggests that charge-based molecular interaction may be associated with the C3–Cp40 interaction in the zebrafish molecule. It has been shown that charged residues in Cp40 play important roles in molecular interaction [63]. In the present study, the measurement of the third binding force based on hydrogen bonds showed that the network of hydrogen bonds in *D. rerio* was more abundant than in *H. sapiens*, especially at the Cp40 binding site between the MG4 and MG5 domains.

Cavitation analysis on C5aR1 showed the main site for binding with PMX205 in the two species. In zebrafish, the receptor was found to have a pocket with greater volume, which probably conferred better ability to accommodate the inhibitor (Figure 6c,d). In the context of molecular bonds, possibly the most relevant of these was the hydrophobic interaction, since the C5aR1 receptor presents an abundant hydrophobic area, especially in *D. rerio*. The distribution of positive and negative charges in C5aR1 in *D. rerio* was less abundant than in the homologous molecule of *H. sapiens*, and this result suggests that, to some extent, the binding of PMX205 to the activity site may also be associated with ionic bonds (Figure 6e,f). The distribution of hydrogen bonding networks was in fact more abundant in the intracellular region of C5aR1, while there were only two networks located in the orthosteric site of the receptor, in both species. Nonetheless, despite the possibility that hydrogen bonding is correlated with PMX205–C5aR1 interaction, the reduction of these networks at the orthosteric site may mean that other chemical interactions are more closely related to this interaction. Lastly, the combination of these intermolecular binding forces is crucial for peptide–protein interaction, and this acts directly on the energy expenditure of the ligand–receptor interaction.

Continuing with this hypothesis, the peptide–protein interaction was validated through two docking simulations that showed specific and nonspecific bindings for the two interactions of Cp40-C3 with PMX205-C5aR1, for both species in the first screening (Figure 3c). In the second Cp40-C3 screening, in the docking directed by the receptor binding residues, specific centralized bindings could be seen. These resulted in more than 10 models for fitting at the binding site. Furthermore, the magnitudes of the ten interactions were similar between the two species (Figure 3e). Thus, in order to choose the best interaction, the effectiveness of Cp40 binding was evaluated using the docking score (−116.71), confidence score (0.3394) and ligand RMSD (8.22), given that this binding is directly correlated with the location of docking in the active site region, as previously postulated for the 2QKI crystallographic model.

In the literature, the binding region for Cp40 was identified in the C-terminal portion of the β chain of the human C3 molecule, and crystallographic analysis revealed a binding pocket formed by macroglobulin (MG) domains 4 and 5 [64]. Furthermore, the interaction site of Cp40 with the zebrafish C3 molecule was also similar, and was present between residues 578 and 645, inserted in the MG4 and MG5 domains in a similar way to the interaction with the human molecule [80].

In the present study, we also evaluated the interaction of PMX205 with C5aR1 in both zebrafish and humans in silico. The peptide–protein interaction was validated using the neural network. Through screening the interaction based on PMX205–C5aR1 docking between the species, a predominance of nonspecific bindings was observed in both simulations, for *H. sapiens* and *D. rerio* (Figure 7c). However, *D. rerio* presented two specific interactions, whereas *H. sapiens* had only one interaction, located at the binding site. Compared with the result from the second docking on the Hdock platform (Figure 7f), the PMX205–C5aR1 interaction of *H. sapiens* ended up centralizing the 10 models only in the binding site, unlike *D. rerio*, which showed two nonspecific interactions. Despite this, the docking score values for interactions/Models 6 and 7 of *D. rerio* were higher than those of *H. sapiens*. This evidence may suggest that the PMX205–C5aR1 interaction occurs in the biological system of *D. rerio*. The action of PMX205 in the zebrafish model has already been evaluated in vivo, although no data on the interaction site of this molecule with C5aR1 of zebrafish are available. Indeed, PMX205 was able to reduce the expression and activation of C5aR1 after the induction of cardiac lesions in zebrafish, in a model for cardiac regeneration [81]. PMX205 also reduced the migration of osteoblasts in in vivo lesions in the zebrafish model [82].

The validation of virtual screening in silico, using molecular docking and the transposition of the results from this interaction to in vitro and in vivo studies, has become established in the scientific literature. There is much evidence to suggest that biophysical mechanisms in virtual environments are, in fact, replicated in biological systems [83,84,85]. The in silico interaction between the nuclear hormone receptor of *Caenorhabditis elegans* and thirty-three environmental chemical products obtained from the Tox21 database can be highlighted. When transposed to an in vivo toxicity study, clear evidence was found that the molecular interaction effect observed in the virtual environment was somewhat similar to that of the biological model [86]. Furthermore, the transposition of in silico virtual screening of anticancer drugs using the receptors caspase-3, Bcl-2 and TRAF2 and the interaction proteins kinase and cyclin-dependent kinase 2 (CDK2) to an in vitro trial was validated through the in vitro anticancer activity against HCT-116 and the cell line HeLa [87]. In addition, results from molecular docking studies were crucial for validating the interactions of twenty phytocompounds with high binding affinity for the target receptors AChE, COX2 and MMP8, which are involved in the physiopathology of Alzheimer’s disease. When transposed to an in vivo trial, these phytocompounds showed potent neuroprotector effects that directed studies towards a preclinical phase with monotherapy or combined therapy for Alzheimer’s disease [88].

In close harmony with animal welfare, bioinformatic tools enable a precise assessment of the capacity of a substance for use as a drug and elucidate the possible targets and therapeutic molecules before these are tested in vitro or in vivo [89]. Moreover, the predictive power of the virtual screening of molecular interactions through the use of artificial intelligence has enabled important advances towards a comprehension of pathological processes, such as in relation to drugs with hepatotoxic potential [90], screening for molecules with deleterious effects on mitochondria [91] and, especially, the discovery of drugs and vaccines against newly emerging diseases such as the pandemic caused by the new coronavirus [92,93,94], and the development of vaccines against other pathogens, such as *Moraxella catarrhalis* [95], *Acinetobacter baumannii* [96] and lymphocytic choriomeningitis virus [97].

In conclusion, given the robustness of bioinformatic tools, we can suggest that the results from our study bring a new perspective to the use of zebrafish. Not only is *D. rerio* a model for studying the complement system but also it enables the evaluation of new drugs that act on the complement system pathways. The in silico test on PMX205 demonstrated a pattern of reverse engineering, such that the effectiveness of PMX205 in zebrafish was proven and revalidated through the virtual test. The test showed the molecular interaction in greater detail and confirmed the connection between PMX205 and C5aR1. In addition, we suggest that Cp40 has a promising effect with regard to blocking the cleavage of C3 into C3b in the complement system of zebrafish, in the same way that this occurs in the homologous human molecule, due to the characteristic strength of the affinity of the peptide–protein binding. However, despite these promising results, additional tests are still needed in order to analyze the molecular dynamics more accurately and make correlations with biophysical tests. In this initial analysis, our aim was merely to define the possibility of peptide–protein interaction and to demonstrate that zebrafish form a representative biological system for therapeutic studies on disorders of the complement system. Therefore, further in vitro and in vivo studies are necessary, with the aims of clearly defining the transposition of the virtual effect to the biological system and defining the interactions of Cp40 and PMX205, through molecular assays using resonance and other biophysical methods, so as to ascertain the molecular dynamics in greater detail.

## 4. Materials and Methods

### 4.1. FASTA Sequences and Similarity Analysis on C3 and C5aR1 between H. sapiens and D. rerio

The amino acid sequences in the FASTA format of C3 and C5aR1 from *D. rerio* and *H. sapiens* were acquired from the UniProt database (https://www.uniprot.org/), under the identification codes Q3MU74 for *D. rerio* C3 and C5AR1_DANRE for *D. rerio* C5aR1, and P01024 for *H. sapiens* C3 and C5AR1_HUMAN for *H. sapiens* C5aR1. The percentage similarity between homologous proteins of the two species was calculated using the EMBOSS Water platform (https://www.ebi.ac.uk or https://www.ebi.ac.uk/Tools/psa/emboss_water/) and the alignment results generated as ALN files were entered into the ESPript platform (http://espript.ibcp.fr). The settings were designed to detect secondary structures, using the PDB file (PDB ID: 2QKI) of human C3 as a template for the alignment and detection of *D. rerio* C3 secondary structures. Next, we evaluated and compared the proteoforms/subcellular localization of *H. sapiens* and *D. rerio* proteins (C3 and C5aR1) and visualized the proteoform through Protter v. 1.0 (http://wlab.ethz.ch/protter/start/) using FASTA archives.

### 4.2. Preparation of D. rerio and H. sapiens C3 and C5aR1 Structures and Coordinated Files for Docking

We used the PDB ID: 2QKI model acquired from the protein database (https://www.rcsb.org/) to obtain the structure of the β chain of *H. sapiens* C3 protein. For virtual construction of the 3D structure of the *D. rerio* C3 homologous protein, the FASTA file (Q3MU74_DANRE) containing the amino acid sequences was downloaded. Subsequently, virtual prediction of the structure in 3D form was performed on the SWISS_MODEL server (https://swissmodel.expasy.org/) and the PDB file was processed. The model with the highest identity with the reference molecule was acquired. Next, the C3 PDB files of the two species were edited in PyMOL 2.5 software (https://pymol.org/2/). In the 2QKI crystallographic model of *H. sapiens*, obtained from the database, the water molecules and solvents present were edited and deleted in PyMOL, so as not to interfere with the docking analysis. For the docking-based interaction, we edited the C3 of both species and constructed only the β chain, known just by its six domains (MG1-6). Lastly, we inserted the PDB files in the GalaxyWeb server (https://galaxy.seoklab.org/) to refine the structures, based on the refinement of the loop or terminal regions through modeling.

The same procedure was performed to obtain the C5aR1 structures from *D. rerio* and *H. sapiens*. To construct *H. sapiens* structures, the crystallographic model 6C1R and the FASTA sequence with identification based on Uniprot (www.uniprot.org) with C5AR1_HUMAN for humans and C5AR1_DANRE for *D. rerio* were used as the basis. After prediction and choosing the model, PBD files were edited in PyMOL 2.5 software.

### 4.3. Virtual Construction of Cp40 and PMX205

Construction of the Cp40 peptide in PDB format was performed on the PEPstrMOD server (https://webs.iiitd.edu.in/raghava/pepstrmod). For this, the Cp40 amino acid sequence (D-Tyr-Ile-[Cys-Val-(1Me)Trp-Gln-Asp-Trp-Sar-Ala-His-Arg-Cys]-(Me)Ile) was inserted and the server was configured to include isomerisms and methylations. Upon completion of the configurations, the sequence was submitted to the server and the PBD Cp40 file was generated. PMX205 (N(2)-(3-phenylpropanoyl)-L-Orn-L-Pro-3-D-Cha-L-Trp-L-Arg) was obtained in Mol2 2D format from the ChemSpider database (http://www.chemspider.com/), under identification number ID 5294036.

### 4.4. Comparative Analysis of Hydrophobic Areas, Negative and Positive Charges and Hydrogen-Bonding Networks on the Protein Surfaces of C3 and C5aR1

Comparative analysis of the area distribution of positive and negative charges and the hydrophobicity of the proteins was performed on the iCn3D server (www.ncbi.nlm.nih.gov/Structure/icn3d/full.html). For this, the PDB files of the proteins chosen were inserted into the platform. The server then presented the data in 3D format and generated a table containing the number of residues classified according to the configuration of choice. The area of these residues was calculated using the solvent-accessible surface area (SASA) (Å2). For the distribution analysis on hydrogen bond networks, the ProteinTools server (https://proteintools.uni-bayreuth.de/bonds/structure) was used. 

Cavitation was performed only for C5aR1, to generate pockets and compare the homologous structures of *D. rerio* and *H. sapiens*. Data from the PDB file were entered into the Caver Analyst 2.0 software (https://caver.cz/) and the pocket volumes were tabulated. The first five values, relating to the largest pockets, were compared between the species. This analysis was performed only for C5aR1 due to the location of the binding site, in order to compare the volume of pockets inserted in the PMX205 binding site.

### 4.5. Virtual Screening Based on the Docking of Cp40 and PMX205 with D. rerio and H. sapiens C3 and C5aR1 Molecules

For analysis based on docking, two simulations were used. In the first one, the CB-Dock2 server (https://cadd.labshare.cn/) was used to screen the main specific connections close to the region of the C3 interaction site and, in addition, nonspecific links were mapped. These connections were based on cavity detection. Molecular fitting was performed based on AutoDock Vina and then a blind docking procedure was performed based on homologous models. All of these parameters were provided by the CB-Dock2 server. Subsequently, to configure the server, the PDB files of the receiver and the ligands of both species were inserted. Considering that the model molecule information (PDB ID: 2QKI) accurately provides the location of the binding site of C3-compstatin, the PDB file for the analysis on docking and homologous template fitting was added. The results from this generated ten interactions. Then, with the results obtained from CB-Dock2, the main peptide–protein interactions detected were determined on the surface or close to the binding site fitting, taking the model (PDB ID: 2QKI) as the basis. The results from this first interaction made it possible to choose the ideal model based on the location and separation of the main receptor residues that interact with the ligand. Amino acids were then selected in the following positions: 369:A, 420:A, 421:A, 422:A, 423:A, 450:A, 451:A, 452:A, 453:A, 455:A, 456:A, 457:A, 458:A, 459:A, 460:A, 484:A, 485:A, 486:A, 489:A, 511:A, 513:A, 515:A.

In the second simulation of the docking analysis, the β chains of C3 and Cp40 were uploaded and the HDOCK SERVER server (http://hdock.phys.hust.edu.cn/) was configured through the insertion of the positions of the amino acids mentioned above. The simulations were restricted to the cavity where the binding site was located. The compilation of these data generated ten possible models and these results were presented in 3D and according to the values of the docking tables. Finally, the most promising model was selected based on its location in relation to the binding site and in correlation with the docking value. This redocking model was analyzed on the FlexPepDock server (lexpepdock.furmanlab.cs.huji.ac.il) and the 3D model of choice was defined through comparison with the peptide–protein interaction model (PDB ID: 2QKI).

For the interaction of PMX205 with C5aR1 in both species, the same pathways as for the C3–Cp40 interaction were used. Screening was performed using CB-Dock2 to detect possible specific and non-specific binding, based on cavitation, blind docking and pairing in the well-resolved model (6C1R). The sequence of the amino acid residues of the receptors located in the regions was the following: 71:A, 84:A, 85:A, 87:A, 88:A, 142:A, 146:A, 149:A, 159:A, 160:A, 161:A, 162:A, 167:A, 170:A, 171:A, 174:A, 175:A, 177:A, 178:A, 229:A, 230:A, 231:A, 233:A, 234:A, 236:A, 237:A, 239:A, 241:A, 246:A, 250:A, 253:A, 254:A, 257:A. This was entered in the HDOCK SERVER configuration and new docking was simulated.

Then, after determining the best interaction for both the C3-Cp40 and the C5aR1–PMX205 complex, the PBD data were edited in PyMOL 2.5 software (https://pymol.org/2/) and were shown in detail, separately from the screening docking.

## Figures and Tables

**Figure 1 ijms-24-13895-f001:**
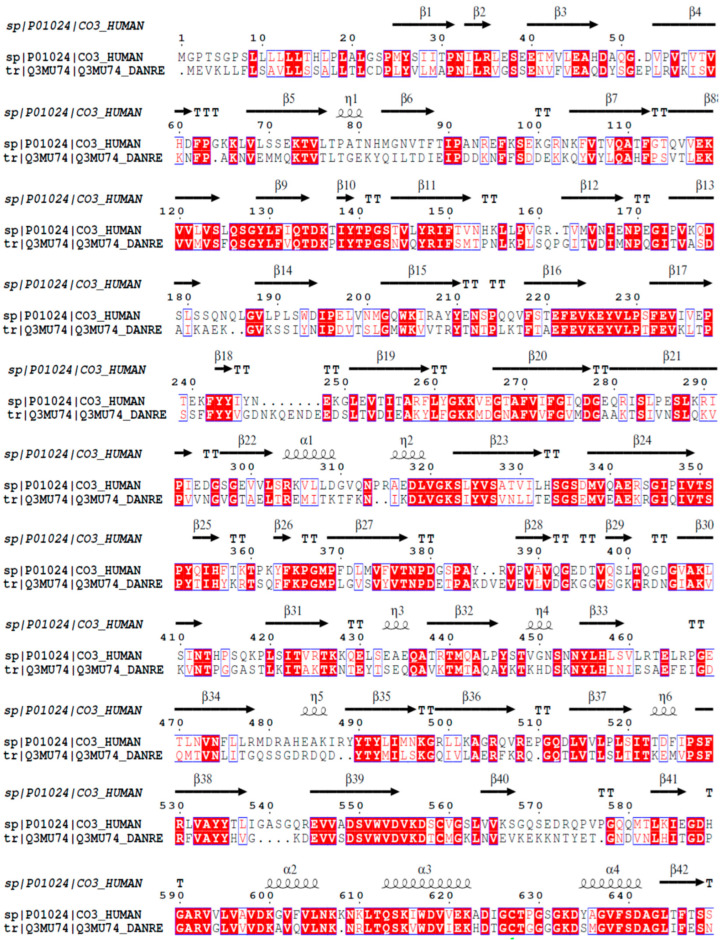
Homology analysis between the β chains of C3 molecules from *H. sapiens* and *D. rerio*. The FASTA sequences CO3_HUMAN and Q3MU74_DANRE were obtained from the UniProt database (http://www.uniprot.org), and their similarity and identity were evaluated on the Espript platform (http://espript.ibcp.fr/ESPript/cgi-bin/ESPript.cgi). Similar amino acid sequences in the two species are shown in red, and the similarity of the secondary structures is shown by the symbols β, α and η.

**Figure 2 ijms-24-13895-f002:**
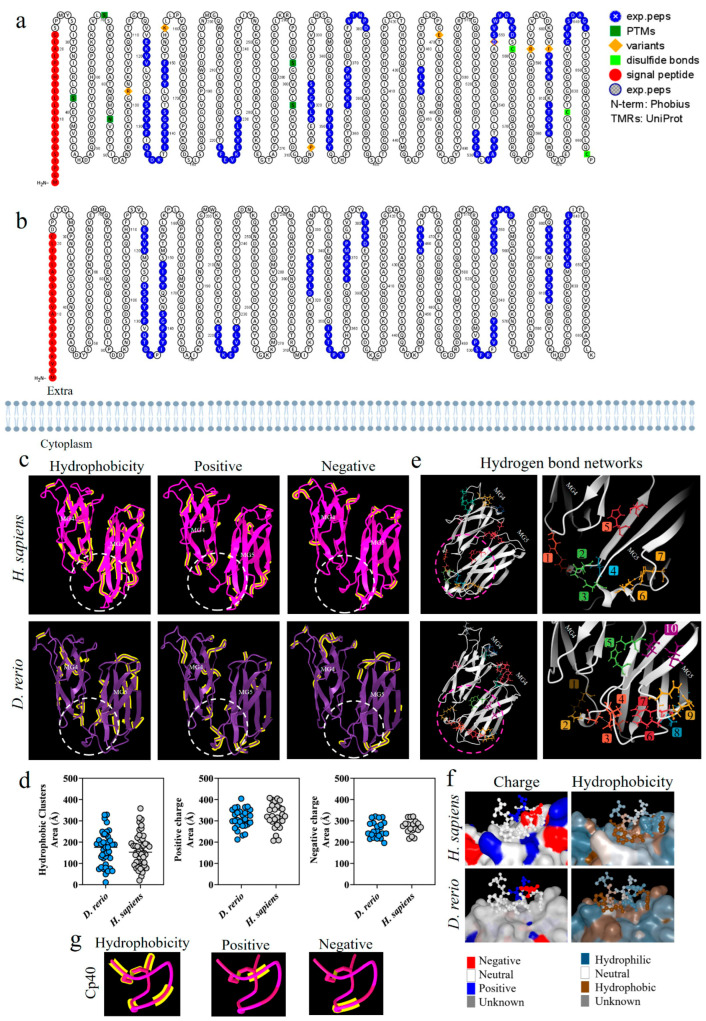
Comparative analysis on the C3-molecule β chains from *D. rerio* and *H. sapiens*. (**a**,**b**) Representation of C3 proteoforms (residues 1-645) from *D. rerio* and *H. sapiens* showing similar peptide sequences between species, as identified in blue; (**c**) structural and physicochemical characteristics of the C3 β-chain MG4 and MG5 domains in three-dimensional form showing the properties of amino acid residues in terms of hydrophobicity and positive and negative charges; (**d**) graphs showing the distribution of these amino acids according to area in Å; the networks are numbered and correlated with the color of the residuals; (**e**) distribution of hydrogen bond networks in the MG4 and MG5 domains; the circle drawn shows in detail the regions of the binding site based on the PDB model: 2QKI. In (**f**), the distribution of areas according to charges and hydrophobicity, on the surface between the ligand and the receptor at the binding site, is shown in detail, and in (**g**) the charge and hydrophobicity properties of Cp40 are exhibited. The numerical sequences refer to the colors that represent each network of hydrogen bonds.

**Figure 3 ijms-24-13895-f003:**
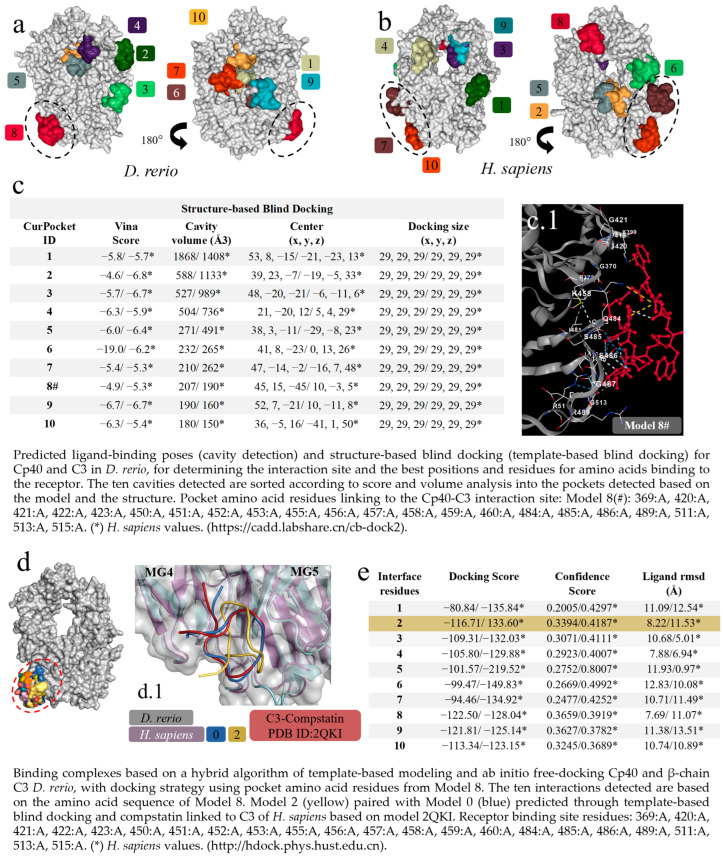
Virtual screening for the detection of the best *D. rerio* Cp40–C3 interaction. (**a**) Docking shows the specific interaction (Model 8, in red) between Cp40 and the active site, as well as between the MG4 and MG5 domains of the *D. rerio* C3 β chain. Other interactions/models outside the circle are unspecific links; (**b**) *H. sapiens* C3 β chain docking showing two model-specific interactions (7 and 10) and other non-specific bindings; (**c**) shows the top ten interactions based on cavitation and blind docking; (*) represents *H. sapiens* values and (#) shows model 8#, selected according to position associated with the Cp40 activity site and the compstatin 2QKI crystallographic model. (**c.1**) Interaction of Cp40 with the main C3 binding amino acid residues in the orthosteric binding pocket between the MG4 and MG5 domains of Model 8#. (**d**) Docking based on the amino acid sequence of Model 8 (**d.1**) shows three interactions: model 0 (blue), based on prediction according to template-based blind docking; compstatin linked to C3 of *H. sapiens* (red), based on model 2QKI; and model 2 (yellow), chosen from the positions and values shown in the (**e**). The chains overlap between *H. sapiens* (purple) and *D. rerio* (grey), and the MG4 and MG5 domains distinguish the two domains. The numeric sequences refer to the colors that represent each docking interaction model.

**Figure 4 ijms-24-13895-f004:**
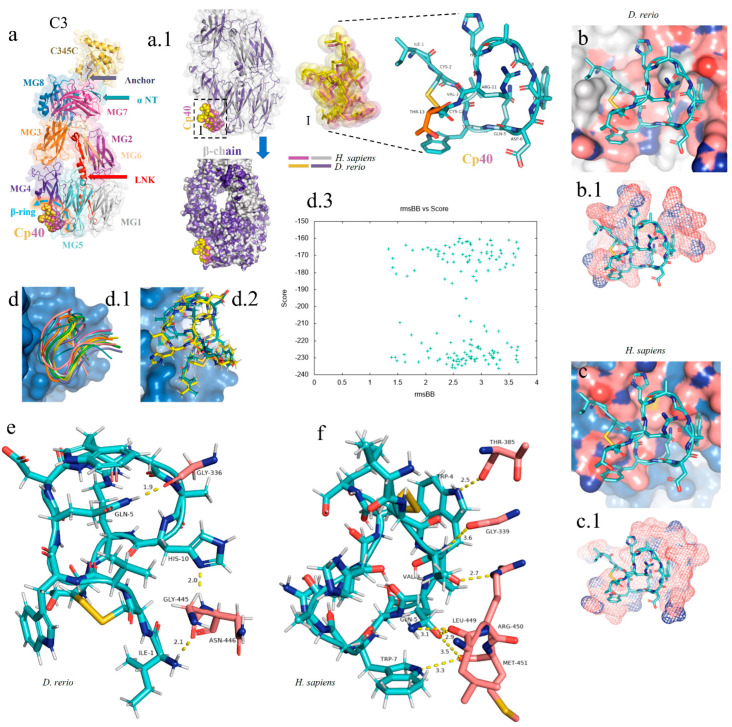
In silico analysis of the peptide–protein interaction between the peptide Cp40 and the C3 molecule of *D. rerio* and *H. sapiens*. (**a**,**a.1**) Structure of the C3/Cp40 complex of *H. sapiens* and the structural alignment of the β chain of the homologous molecule of *D. rerio* (purple); (**I**) comparison of the binding of Cp40 to the domains MG4 and MG5 of the β ring in *H. sapiens* (purple) and *D. rerio* (yellow); (**b**,**c**) docking of the protein–binder interaction between C3 and Cp40 in the two species; (**b.1**,**c.1**) spatial interaction of Cp40 at the binding site; (**d**) redocking, showing the flexibility of the peptide Cp40 at the C3 binding site of *D. rerio*; (**d.1**,**d.2**) the two main interactions with greatest binding force, obtained through the FlexPepDock platform; (**d.3**) graph of RMSD (*x*-axis) vs. score (*y*-axis) of the ten models created through simulations; (**e**,**f**) main interactions of amino acid residues from the pocket of the β chain and the Van der Waals binding force, with binder residues: (**e**) *D. rerio* and (**f**) *H. sapiens*. The PDB C3 file for *D. rerio* was built in the Swiss model, using the reference 5fo8 of *H. sapiens*, and the Hdock Serve platform for docking. Macroglobulin domain (MG); binding domain (LNK); cleavage site of the alpha chain (NT); anchor (hydrogen bridge binding to the MG7 domain).

**Figure 5 ijms-24-13895-f005:**
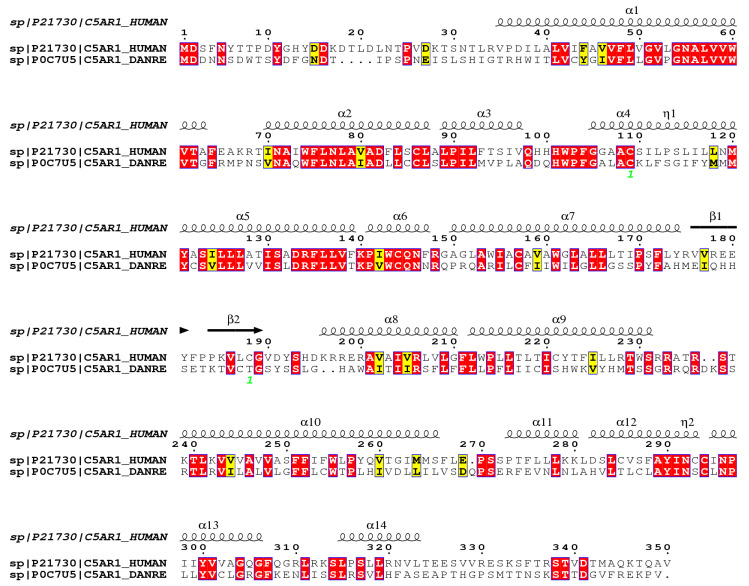
Comparative structural alignment between *D. rerio* and *H. sapiens* C5aR1. The FASTA sequences C5AR1_HUMAN and C5AR1_DANRE were obtained from the UNIPROT database (http://www.uniprot.org), and their similarity and identity were analyzed on the Espript platform (http://espript.ibcp.fr/ESPript/cgi-bin/ESPript.cgi). Similar amino acid sequences in the two species are shown in red, and the similarity of the secondary structure is shown by the symbols β, α and η.

**Figure 6 ijms-24-13895-f006:**
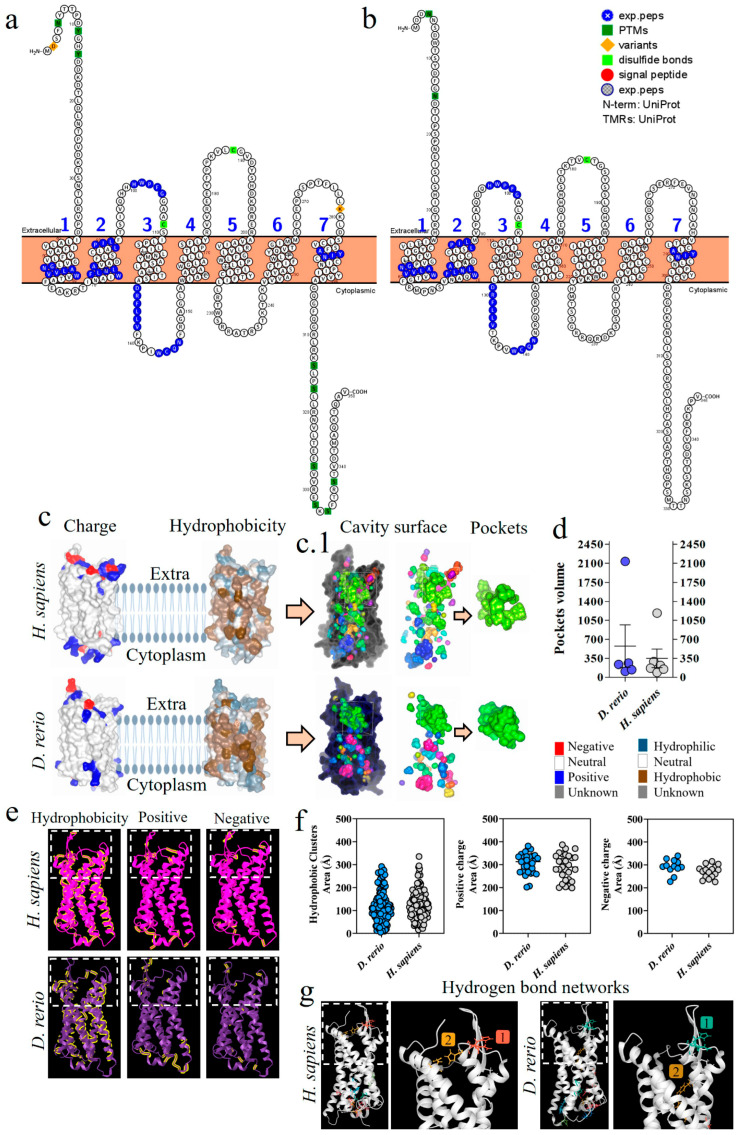
Comparative analysis of C5aR1 from *D. rerio* and *H. sapiens*. (**a**,**b**) C5aR1 proteoforms from *D. rerio* and *H. sapiens* showing similar peptide sequences identified in blue; (**c**) structural and physicochemical characteristics of C5aR1, showing charge and hydrophobicity; (**c.1**) arrows point to the C5aR1 cavitation and show the PMX205 receptor binding pockets and (**d**) the volume of the pockets; (**e**) structural and physicochemical characteristics of C5aR1 in three-dimensional form, showing the properties of amino acid residues in terms of hydrophobicity and positive and negative charges; the graphs (**f**) show the distribution of these amino acids according to the area in Å; (**g**) distribution of hydrogen bond networks. The numerical sequences refer to the colors that represent each network of hydrogen bonds.

**Figure 7 ijms-24-13895-f007:**
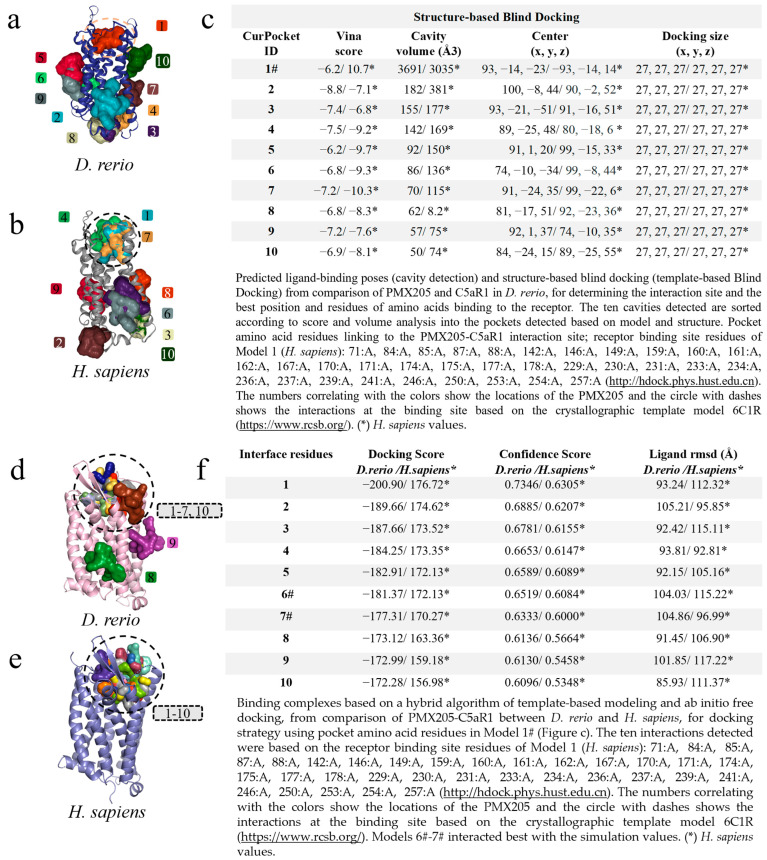
Virtual screening for the detection of the best *D. rerio* PMX205–C5aR1 interaction. (**a**) Docking showing specific interactions inside the drawn circle and non-specific ones outside the circle; (**b**) anchoring of PMX205-C5aR1 from *H. sapiens* showing model 1# specific interaction and other non-specific binding; (**c**) shows the top ten transients with regard to cavitation and blind docking; (*) *H. sapiens* values and (#) Model 1 selected through the position associated with PMX205 activity site based on crystallographic model 6C1R; (**d**) docking based on the amino acid sequence of Model 1# (Figure (**c**)), showing eight specific interactions inside the circle and two outside the circle representing nonspecific simulations from *D. rerio*; (**e**) for *H. sapiens*, all interactions were specific; (**f**) shows the values used for choosing the best interaction. (*) *H. sapiens* values. The numeric sequences refer to the colors that represent each docking interaction model.

**Figure 8 ijms-24-13895-f008:**
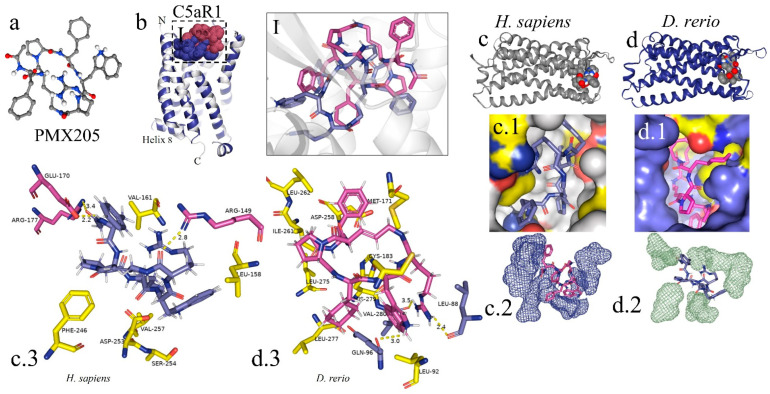
In silico analysis of the interaction of PMX205 with C5aR1 in *D. rerio* and *H. sapiens*. (**a**) 3D structure of PMX205 in the mol2 format of the ChemSpider database; (**b**) structural overlap of *H. sapiens* C5aR1 (grey) and homologous molecule of *D. rerio* (blue); (**I**) comparison of the PMX205–C5aR1 interaction between *H. sapiens* (purple) and *D. rerio* (pink); (**c**,**c.1**,**d**,**d.1**) docking of protein–binder interaction between C5aR1 and PMX205; (**c.2**,**d.2**) interaction of PMX205 with the binding site, in detail; (**c.3**,**d.3**) main amino acid residues of the receptors of *H. sapiens* (**c.3**) and *D. rerio* (**d.3**) and Van der Waals binding force with binder residues. The PDB C5aR1 file for *D. rerio* was built in the Swiss model, using the reference based on the crystallographic model 6C1R of *H. sapiens*, and the CB-Dock platform for docking.

## Data Availability

Not applicable.

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
