# Peer review of "Complement System Inhibitory Drugs in a Zebrafish (Danio rerio) Model: Computational Modeling"

_ijms, 2023, doi:10.3390/ijms241813895_

Round 1

Reviewer 1 Report (Previous Reviewer 2)

My previous comments have been addressed.

English writing should be minor checked.

Author Response

Dear Reviewer,

Thank you for reviewing our manuscript. The revised version of our manuscript was corrected by a professional translator of english. The modifications are highlighted in blue. 

Best regards, 

Denise

Reviewer 2 Report (New Reviewer)

This work addresses the potential use of the zebrafish as a robust animal model for studying the complement system using molecular docking interactions between peptides and targets. The idea is interesting and is supported by standard bioinformatics methods. However, the validity of the results would be better if the docking predictions were supported by MD simulations that provide insights into the energetics and stability of binding. The authors should at least discuss and comment on the possibility of further computational validation of their results.

The paper is well written and the results are presented in an understandable way, so I recommend publication in IJMS after a minor revision.

Minor editing of English language required

Author Response

Dear Reviewer,

Thank you for reviewing our manuscript. We add in blue highlights in the text the future tests using molecular dynamics to be carried out.

We would like to clarify about the absence of the molecular dynamics test. In this first manuscript, our hypothesis was limited by the possibility of peptide-protein interaction between the complement inhibitors Cp40 and PMX205 with the zebrafish target molecules.

That said, at the moment, we are carrying out the biophysical experiments, we intend to characterize the zebrafish C3 and C5aR1 by crystallography and carry out the surface plasmon resonance interaction analysis tests and, together with these new data, correlate with the computational analyses. Thus, we chose to leave the analysis of molecular dynamics for the second experimental moment, in order not to conflict with the first experimental phase, which is limited to the docking interaction and to be complementary with the biophysical analyzes in the second manuscript.

We are working on fine-tuning the experiments and moving towards in vivo testing as we work on the biophysical experiments.

Best regards, 

Denise

This manuscript is a resubmission of an earlier submission. The following is a list of the peer review reports and author responses from that submission.

Round 1

Reviewer 1 Report

The manuscript is well written and interesting. The methods are clearly presented, in addition to a detailed and well-explained presentation of results resulting from them. Conclusions are well supported by the demonstrated results. There are a couple of minor comments I have:

1. The goal of the manuscript, as well as its novelty should be better addressed by the authors in the introduction. I would suggest adding an additional paragraph further detailing these elements.

2. Figure 7 - there is an extra dot in the caption

3. Figure 5 - there seems to be a line cut off (l. 174) by the figure.

Due to the above comments, I suggest minor revisions to the manuscript prior to the publication.

Author Response

Question 1. The goal of the manuscript, as well as its novelty should be better addressed by the authors in the introduction. I would suggest adding an additional paragraph further detailing these elements.

Response: Dear reviewer, thank you for reviewing our manuscript. We inserted new snippets into the objectives as suggested.

Question 2. Figure 7 - there is an extra dot in the caption

Response: Corrected.

Question 3. Figure 5 - there seems to be a line cut off (l. 174) by the figure.

Response: Corrected.

Reviewer 2 Report

In this study, the authors proposed a Complement system inhibitory drugs in zebrafish (Danio rerio) model. Although the results seem promising, some major points should be addressed as follows:

1. The study provides limited information about the diseases associated with complement activation and the potential therapeutic significance of inhibiting complement pathways. Providing more context and citing relevant literature would help readers understand the importance and relevance of the study.

2. The study mentions "computational modeling" and "molecular docking interactions" but does not provide sufficient details on the specific computational methods and software used. Without this information, it is difficult to assess the validity and reliability of the in silico analysis.

3. The study briefly mentions "positive interaction" between the peptides and their respective targets in zebrafish but does not elaborate on the significance or implications of these interactions. It is important to provide a more detailed interpretation of the results, including any potential limitations or uncertainties.

4. The study mentions the intention to validate the in silico findings in vivo, but it does not provide any results or discussion on the actual validation experiments. It would be valuable to include the findings from the in vivo validation to support the conclusions drawn from the computational modeling.

5. More references to computational modeling/bioinformatics should be added to attract a broader readership i.e., PMID: 31886259, PMID: 31921391.

6. While the study focuses on the zebrafish model, it does not discuss the potential implications of the findings for other animal models or humans. It would be helpful to address the generalizability of the results and their potential relevance to therapeutic studies in other species.

7. Overall, English writing and presentation style should be improved.

Overall, English writing and presentation style should be improved.

Author Response

Question 1. The study provides limited information about the diseases associated with complement activation and the potential therapeutic significance of inhibiting complement pathways. Providing more context and citing relevant literature would help readers understand the importance and relevance of the study.

Response: Dear reviewer, thank you for reviewing our manuscript. We have inserted new paragraphs in the introduction as suggested.

Question 2. The study mentions "computational modeling" and "molecular docking interactions" but does not provide sufficient details on the specific computational methods and software used. Without this information, it is difficult to assess the validity and reliability of the in silico analysis.

Response: Dear reviewer, we presented these computational methods and softwares in two ways (i): describing them in the topic materials and methods and (ii) by a flowchart (Figure 7) showing in detail the platforms and softwares used for the in silico analysis.

Since we have used computational tools that already exist and are well used in the literature (Bordoli et al., 2009; Heo et al., 2013; Omasits et al., 2014; Yan et al., 2020), we can assume that our in silico analysis is valid. We have used available tools already stablished in the bioinformatics science.

Bordoli, L.; Kiefer, F.; Arnold, K. et al. Protein structure homology modeling using SWISS-MODEL workspace. Nature Protocols 2009, 4, 1–13. https://doi.org/10.1038/nprot.2008.197

Heo, L.; Park, H.; Seok, C. GalaxyRefine: Protein structure refinement driven by side-chain repacking. Nucleic Acids Research. 2013, 41, 384-8. https://doi.org/10.1093/nar/gkt458.

Omasits, U.; Ahrens, C.H.; Müller, S.; Wollscheid, B. Protter: interactive protein feature visualization and integration with experimental proteomic data. Bioinformatics. 2014, 30(6), 884-6. https://doi.org/10.1093/bioinformatics/btt607.

Yan, Y.; Tao, H.; He, J.; et al. The HDOCK server for integrated protein–protein docking. Nature Protocols 2020, 15, 1829–1852. https://doi.org/10.1038/s41596-020-0312-x

Question 3. The study briefly mentions "positive interaction" between the peptides and their respective targets in zebrafish but does not elaborate on the significance or implications of these interactions. It is important to provide a more detailed interpretation of the results, including any potential limitations or uncertainties.

Response: Dear Reviewer, thank you for your suggestion and we are very sorry. We should make this information more clear. In fact, the interaction of Cp40 with the zebrafish C3 molecule was the first step to allow us to proceed with the in vivo assays. We are fully aware that virtual screening is limited and the programming process and language are constantly evolving.  Moreover, no drug is approved only by computational analysis, always requiring an in vivo study, as we made clear in the discussion.

Question 4. The study mentions the intention to validate the in silico findings in vivo, but it does not provide any results or discussion on the actual validation experiments. It would be valuable to include the findings from the in vivo validation to support the conclusions drawn from the computational modeling.

Response: Dear reviewer, since this special issue of IJMS deals with the topic "Advanced Research in Prediction of Protein Structure and Function" we decided to address only the in silico study. The in vivo study is still ongoing and this will be the subject of further publications.

Question 5. More references to computational modeling/bioinformatics should be added to attract a broader readership i.e., PMID: 31886259, PMID: 31921391.

Response: Dear reviewer, thank you for your reference suggestions. References have been inserted.

Question 6. While the study focuses on the zebrafish model, it does not discuss the potential implications of the findings for other animal models or humans. It would be helpful to address the generalizability of the results and their potential relevance to therapeutic studies in other species.

Response: Dear reviewer, our research group has as a priority the postulation of  zebrafish as a model to study the complement system and to test complement inhibitors.

According to Lamers et al. (2022) in complement (C) inhibition assays using CP40 and based on antibody-mediated C-activation in plasmas from human, mouse, rat, rabbit, dog or pig, showed strong inhibitory activity of this inhibitor in human plasma, while no notable inhibition of complement activation was observed in non-primate species. Therefore, we have limited this manuscript to virtual screening of the interaction between Cp40 and the C3 molecule from zebrafish, a species that has yet to be tested.

Since PMX205 has already been tested in zebrafish (Natarajan et al., 2018 and Sehring et al., 2022) and indeed the peptide-protein interaction has been confirmed by its pharmacological effect. We used in silico modeling for our further future analysis in vivo. That said, our interest is to use the zebrafish model to understand the pathological aspects involved in diseases associated with the complement system disturbance and, secondly, to test drugs with C-inhibitory potential. Cp40 and PMX205 inhibitors are already being tested in vivo in zebrafish in our group for diseases involving the complement system. This is well described in the discussion.

Lamers, C.; Xue, X.; Smieško, M.; van Son, H.; Wagner, B.; Berger, N.; Sfyroera, G.; Gros, P.; Lambris J.D.; Ricklin, D. Insight into mode-of-action and structural determinants of the compstatin family of clinical complement inhibitors. Nature Communications. 2022, 13(1), 5519. https://doi.org/10.1038/s41467-022-33003-7

Natarajan, N.; Abbas, Y.; Bryant, D.M.; Gonzalez-Rosa, J.M.; Sharpe, M.; Uygur, A.; Cocco-Delgado, L.H.; Ho, N.N.; Gerard, N.P.; Gerard, C.J.; et al. Complement receptor C5aR1 plays an evolutionarily conserved role in successful cardiac regeneration. Circulation 2018, 20, 2152-2165. https://doi.org/10.1161/CIRCULATIONAHA.117.030801

Sehring, I.; Mohammadi, H.F.; Haffner-Luntzer, M.; Ignatius, A.; Huber-Lang, M.; Weidinger, G. Zebrafish fin regeneration involves generic and regeneration-specific osteoblast injury responses. Elife. 2022, 11, e77614. https://doi.org/10.7554/eLife.77614.

Question 7. Overall, English writing and presentation style should be improved.

Response: Dear reviewer, the manuscript has been revised.

Reviewer 3 Report

Title: “Complement system inhibitory drugs in zebrafish (Danio rerio) model: Computational Modeling”

This study aimed to investigate in silico, through computational modeling, the hypothesis of the interaction of these Complement inhibitor peptides with the zebrafish target molecules, for later validation in vivo. In particular, the authors have analyzed the molecular docking interactions between peptides and target molecules. The authors claim that this study showed a positive interaction between the Cp40 and the cyclic peptide PMX205 with their respective targets of zebrafish, suggesting that it can be used as animal model for therapeutic studies using these inhibitors.

General comment: This work is interesting and in general well written. However, some minor points still remain to improve the quality of the whole manuscript.

1) Figure 1. Homology analysis between the β chains of C3 molecules from H. sapiens and D. rerio. FASTA CO3_HUMAN and Q3MU74_DANRE sequence obtained from the UniProt database (http://www.uniprot.org) and similarity and identity was evaluated on the Espript platform (http://espript.ibcp.fr/ESPript/cgi-bin/ESPript.cgi).

* )Please improve, the figure caption and the quality of the figure 1. All the labels should be explained in a clear way.

2) Figure 2. Comparative analysis of the β chains of C3 molecules from D. rerio and H. sapiens. (a-b) Representation of C3 proteoforms from D. rerio and H. sapiens showing similar peptide sequences between species, identified in blue. (c) Structural and physicochemical features of C3 showing the 3D structure and properties of amino acid residues in terms of charge (I) and hydrophobicity (II) classified by the Caver Analyst 2.0 Beta program (http://www.caver.cz /). *Domains MG4 and MG5.

*) Please explain better the 3d structures. What is the importance of the unknown segments ?

3 )Figure 3 and captions 3 should be improved. Some parts of the panel 3 are not clearly readable. Please improve and magnify.

4) Figure 6. See the previous comments for figure 3.

Some improvements of the language are still possible

Author Response

Question 1.  Figure 1. Homology analysis between the β chains of C3 molecules from H. sapiens and D. rerio. FASTA CO3_HUMAN and Q3MU74_DANRE sequence obtained from the UniProt database (http://www.uniprot.org) and similarity and identity was evaluated on the Espript platform (http://espript.ibcp.fr/ESPript/cgi-bin/ESPript.cgi).

* )Please improve, the figure caption and the quality of the figure 1. All the labels should be explained in a clear way.

Response: Dear reviewer, thank you for reviewing our manuscript.  Improved as suggested.  

Question 2. Figure 2. Comparative analysis of the β chains of C3 molecules from D. rerio and H. sapiens. (a-b) Representation of C3 proteoforms from D. rerio and H. sapiens showing similar peptide sequences between species, identified in blue. (c) Structural and physicochemical features of C3 showing the 3D structure and properties of amino acid residues in terms of charge (I) and hydrophobicity (II) classified by the Caver Analyst 2.0 Beta program (http://www.caver.cz /). *Domains MG4 and MG5.

*) Please explain better the 3d structures. What is the importance of the unknown segments?

Response: Modified as suggested.  

Question 3. Figure 3 and captions 3 should be improved. Some parts of the panel 3 are not clearly readable. Please improve and magnify.

Response: Modified as suggested.  

Question 4. Figure 6. See the previous comments for figure 3.

Response: Modified as suggested.  

Reviewer 4 Report

Title: Complement system inhibitory drugs in…

Manuscript ID: ijms-2437580

Authors: Fernandes and Tambourgi

Dear Authors,

Thank you for the opportunity to read your article. I found the topic is interesting and fundamental. Generally speaking, the methods and results need more clear explanation and detail discussion with fair point of view. Your statements/discussions are often not based on your results. I suggest that this article will be revised extensively before its re-submission for another review process if applicable. As a conclusion, I recommend its major revision at this state.

I hope my comments are helpful.

Good luck,

A reviewer

Major concerns:

“Keywords”

->Please consider providing keywords that are not used in the article title.

“2. Results”

-In general, please consider describing and discussing your results detail enough for someone else to better understand them. You may find some detail comments below.

2.1. Analysis of the homology of the C3 molecules of Danio rerio and Homo sapiens

-Lines 85-88: “Figure 3 shows…”->Figure 1 shows…?

2.2. Comparative structural analysis on…

-Lines 104-106: “…neutral charge predominates across the surface of the two proteins…”->In this section, please consider discussing the distribution of negatively and positively charged areas/sites. If there is any interaction due to charge, it can be based on the charged surface sites for their repulsion or attraction.

-Lines 107-108: “…hydrophobicity is heterogeneously distributed…while hydrophilic areas predominate…”->In this section, please consider discussing the reason behind.

-In this section, please consider discussing the relationship between the molecules and charge distribution and/or hydrophobicity/hydrophilicity distribution using Fig.2a, b, d.

2.3. In silico analysis of the interaction of…

-Figure 3(b): What are the “ten possible interactions”?

“3. Discussion”

-Please consider citing your results and figures in this section, and linking your results with discussion. The current discussion appears to be isolated from your results.

“4. Materials and Methods”

-In general, please consider describing your methods in detail enough for someone else to conduct the same/similar experiment in the future.

Minor editing of English language required.

Author Response

Question 1. “Keywords”

->Please consider providing keywords that are not used in the article title.

Response: Dear reviewer, thank you for reviewing our manuscript. Keywords have been modified as suggested.

Question 2. “Results”

-In general, please consider describing and discussing your results detail enough for someone else to better understand them. You may find some detail comments below.

2.1. Analysis of the homology of the C3 molecules of Danio rerio and Homo sapiens

-Lines 85-88: “Figure 3 shows…”->Figure 1 shows…?

Response: Modified as suggested.  

-Linhas 85-88: “A Figura 3 mostra…”->A Figura 1 mostra…?

Response: Modified as suggested.  

“2.2. Comparative structural analysis on…”

-Lines 104-106: “…neutral charge predominates across the surface of the two proteins…”->In this section, please consider discussing the distribution of negatively and positively charged areas/sites. If there is any interaction due to charge, it can be based on the charged surface sites for their repulsion or attraction.

Response: Modified as suggested.  

  -Lines 107-108: “…hydrophobicity is heterogeneously distributed…while hydrophilic areas predominate…”->In this section, please consider discussing the reason behind.

-In this section, please consider discussing the relationship between the molecules and charge distribution and/or hydrophobicity/hydrophilicity distribution using Fig.2a, b, d.

Response: Dear reviewer, thank you for your suggestion. However, our interest is to know the similarity of the region of the activity site of Cp40 and PMX205 already postulated in the literature (Jenssen et al., 2007 and Kumar et al., 2020). In this context, on the effect of charges and hydrophobicity on the peptide-protein interaction the binding forces both by charge, hydrophobicity, and hydrogen bonds were computed by the docking interaction and the results of the interaction forces are shown in the graph of the binding free energy of the complex (BFEC) (Figure 3b). We chose to discuss the docking results because their computational data are extremely reliable. Unfortunately, we did not have tools to evaluate the interaction by charge and hydrophobicity separately. Our interest is the similarity of linker amino acid residues between species and the distribution of hydrogen bonds in the molecular interaction.

Janssen, B.J.; Halff, E.F.; Lambris, J.D.; Gros, P. Structure of compstatin in complex with complement component C3c reveals a new mechanism of complement inhibition. J Biol Chem. 2007, 282(40), 29241-7. https://10.1074/jbc.M704587200

Kumar, V.; Lee, J.D.; Clark, R.J.; Noakes, P.G.; Taylor, S.M.; Woodruff, T.M. Preclinical pharmacokinetics of complement C5a receptor antagonists PMX53 and PMX205 in mice. ACS Omega 2020, 5(5), 2345-2354. https://doi.org/10.1021/acsomega.9b03735

“2.3. In silico analysis of the interaction of…”

-Figure 3(b): What are the “ten possible interactions”?

Response: Dear reviewer, thank you for your suggestion. Out of the ten interactions, we have made our choice by the Score and by the interaction that was similar to the interaction site of the human homologous molecule. The interaction of Cp40 with the human C3 molecule occurs in the beta chain (Figure 3e.3). In this way, we structurally aligned human and zebrafish C3 proteins both associated with Cp40 to locate structural overlap and activity site similarity between the two species.

Question 3. “Discussion”

-Please consider citing your results and figures in this section, and linking your results with discussion. The current discussion appears to be isolated from your results.

Response: Included as suggested.

Question 4. “Materials and Methods”

-In general, please consider describing your methods in detail enough for someone else to conduct the same/similar experiment in the future.

Response: Improved as suggested.

Round 2

Reviewer 2 Report

My previous comments have been addressed.

Reviewer 4 Report

Title: Complement system inhibitory drugs in…

Manuscript ID: ijms-2437580

Authors: Fernandes and Tambourgi

Dear Authors,

Thank you for your effort to address my comments. I found the important comments that are necessary for a reader to understand your article were not addressed. I suggest that this article will be revised extensively before its re-submission for another review process if applicable. As a conclusion, I recommend its rejection at this state.

I hope my comments are helpful.

Good luck,

A reviewer

Major concerns:

“2. Results”

2.2. Comparative structural analysis on…

-Lines 117-119: “…neutral charge predominates across the surface of the two proteins…”->In this section, please consider discussing the distribution of negatively and positively charged areas/sites. If there is any interaction due to charge, it can be based on the charged surface sites for their repulsion or attraction.

-Lines 119-121: “…hydrophobicity is heterogeneously distributed…while hydrophilic areas predominate…”->In this section, please consider discussing the reason behind.

-In this section, please consider discussing the relationship between the molecules and charge distribution and/or hydrophobicity/hydrophilicity distribution using Fig.2a, b, d.

Your answer “…our interest is to know the similarity of the region of activity site…the effect of charges and hydrophobicity…were computed…we did not have tools to evaluate the interaction by charge and hydrophobicity separately. Our interest is the similarity of…” is worth mentioned in the revised article. Please consider giving your statements in the revised article in order to clarify what you can do and what you cannot do, and providing your clear interpretation of the results.

2.3. In silico analysis of the interaction of…

-Figure 3(b): What are the “ten possible interactions”? Without their description, a reader cannot understand the meaning and value of your figure. Your reply “out of the ten interactions…” do not answer this question.

“3. Discussion”

-Please consider citing your results and figures shown in the results section of this article (not from literature) in this section, and linking your results with discussion. The discussion still appears to be isolated from your results.

“4. Materials and Methods”

-In general, please consider describing your methods in detail enough for someone else to conduct the same/similar experiment in the future. For example, can your future colleagues perform a follow-up study by reading this paper?

Minor editing of English language required.

Author Response

Dear Reviewer, 

We would like to thank your precious contribution to our manuscript.

We have tried to adjust all our manuscript following your suggestions.

I decided not to answer here your queries, since the volume of modifications was quite big. Thus, please find the modifications (highlighted in blue) in the revised version of the manuscript.

Best regards,

Denise Tambourgi